# Exploration of Synergistic Regulation Mechanisms of Cerebral Ganglion and Muscle in *Eriocheir sinensis* Activated in Response to Alkalinity Stress

**DOI:** 10.3390/ani14162374

**Published:** 2024-08-16

**Authors:** Meiyao Wang, Jun Zhou, Jiachun Ge, Yongkai Tang, Gangchun Xu

**Affiliations:** 1Key Laboratory of Freshwater Fisheries and Germplasm Resources Utilization, Ministry of Agriculture and Rural Affairs, Freshwater Fisheries Research Center, Chinese Academy of Fishery Sciences, Wuxi 214081, China; wangmy@ffrc.cn; 2Wuxi Fisheries College, Nanjing Agricultural University, Wuxi 214081, China; 3Freshwater Fisheries Research Institute of Jiangsu Province, Nanjing 210017, China; finedrizzle@163.com (J.Z.); gjc09@sina.com (J.G.)

**Keywords:** *Eriocheir sinensis*, alkalinity, cerebral ganglion, proteome, metabolome

## Abstract

**Simple Summary:**

It is of great significance to explore the synergistic mechanisms of the brain and muscle in *Eriocheir sinensis* in response to environmental stress. In the present study, proteomics, metabolomics, and combination analyses of the cerebral ganglion and muscle of *Eriocheir sinensis* under alkalinity stress were performed. The results indicated that the cerebral ganglion and muscle played a significant synergistic regulatory role in alkalinity adaptation. This study provides a theoretical reference for further research on the mechanisms of the regulation of the growth and development of *E. sinensis* in saline–alkaline environments.

**Abstract:**

The cerebral ganglion and muscle are important regulatory tissues in *Eriocheir sinensis*. Therefore, it is of great significance to explore their synergistic roles in this organism’s anti-stress response. In this study, proteomics, metabolomics, and combination analyses of the cerebral ganglion and muscle of *E. sinensis* under alkalinity stress were performed. The cerebral ganglion and muscle played a significant synergistic regulatory role in alkalinity adaptation. The key regulatory pathways involved were amino acid metabolism, energy metabolism, signal transduction, and the organismal system. They also played a modulatory role in the TCA cycle, nerve signal transduction, immune response, homeostasis maintenance, and ion channel function. In conclusion, the present study provides a theoretical reference for further research on the mechanisms regulating the growth and development of *E. sinensis* in saline–alkaline environments. In addition, it provides theoretical guidelines for promoting the vigorous development of the *E. sinensis* breeding industry in saline–alkaline environments in the future.

## 1. Introduction

With the continuous intensification of the greenhouse effect, increasing CO_2_ levels are leading to the continuous acidification of oceans and other water resources. This increase in acidification dissolves calcium carbonate, which is an important constituent of the bones and shells of marine organisms [1,2]. It has been reported that the increasing acidity of water has resulted in the formation of thinner cuticles and that it can even cause the extinction of crabs [3,4]. According to a report from the Food and Agriculture Organization of the United Nations (FAO), in 2021, the global area of salt-affected soils covered 424 million hectares of topsoil and 833 million hectares of subsoil [5]. The development and utilization of saline–alkaline land has become a worldwide issue. Research entitled “Carbonization Process and Global Climate Change in saline-alkaline land of Arid Region” indicated that saline–alkaline land can absorb carbon dioxide [6,7,8]. The Chinese mitten crab (*Eriocheir sinensis*), with its tender meat that has made it a delicacy, is an important economic aquatic species. The optimal pH for culturing this crab is above seven, which is alkaline [9,10]. In addition, it has been reported that the symbiotic bacteria on the crab’s carapace play a role in nitrogen fixation, which is important for increasing water fertility [11], reflecting the great potential for culturing the Chinese mitten crab in saline–alkaline water to improve land quality. Research on the gill metabolomics of *E. sinensis* under saline–alkaline stress has indicated that some immune regulatory genes are upregulated after stress [12]. Furthermore, it has been shown that the contents of unsaturated fatty acids and amino acids in *E. sinensis* cultured on saline–alkaline land are higher and that the flavor is better than those cultured in common water [13]. 

There have only been a few studies on the regulation mechanisms of crab species in alkaline environments, and these have mainly concentrated on the harmful effects of CO_2_-driven acidification on crabs and the influence of alkalinity stress on crabs’ physiological metabolism, including the gut microbiota and meat quality [13,14,15,16,17,18,19,20,21,22]. In the horseshoe crab (*Tachypleus tridentatus*) and the mud crab (*Scylla serrata*), CO_2_-driven acidosis can significantly reduce the survival rate, feed intake, growth rate, tissue biochemical indices, and alkaline phosphatase activity, which causes oxidative stress and adverse effects on immune defense function and gut health [21,22]. Under alkaline conditions, Ca^2+^, Na^+^, and K^+^ are the key adaptive ions for mud crabs, and the expression of the AMP pathway is upregulated [20]. The main responsive gut microbiota are Firmicutes, Proteobacteria, Bacteroidetes, and Campilobacteria [18]. Research on the alkalinity response mechanism of *E. sinensis* has suggested that the energy supply is provided through aerobic glycolysis during alkalinity stress [14]. Further research has shown that 96-hour acute saline–alkaline stress causes hepatopancreas damage in *E. sinensis* and enhances the antioxidant system and immunocompetence to resist the stress [15]. By contrast, it has been reported that fattening in saline–alkaline water can improve the color, nutrition, and taste qualities of adult Chinese mitten crabs, while the crude protein contents in the testis and crude fat contents in the muscle of male *E. sinensis* are significantly increased, and the flavor of the hepatopancreas and muscle in male crabs is also improved [13].

Previous studies have indicated that the brain plays a key regulatory role in the release of neurotransmitters, such as serotonin, epinephrine, norepinephrine, and dopamine, as an adaptive response to osmotic stress, such as that caused by salinity and drought [23,24,25]. Existing studies on *Litopenaeus vannamei* have indicated that muscle plays a key modulatory role in the response to osmotic stress [26]. Nevertheless, proteomics and metabolomics, which are very important biological research methods for elucidating scientific problems more accurately and objectively [27,28,29], have not been applied to this system.

In this study, we performed a multi-omics analysis to investigate the regulatory roles of the brain and muscle in the environmental stress response, energy metabolism, and growth and development of *E. sinensis* under acute alkalinity stress. The study revealed the key regulatory proteins in the brain and the key metabolites in muscle under alkalinity stress, as well as the synergistic regulatory response mechanisms in both tissues in resistance to alkalinity stress. The study provides a theoretical reference for the further exploration of the growth and development mechanisms of *E. sinensis* in alkaline environments and provides theoretical guidance for the vigorous development of the *E. sinensis* breeding industry in saline–alkaline environments, as well as the better development and utilization of saline–alkaline land globally.

## 2. Materials and Methods

### 2.1. Experimental Animals and Alkalinity Stress

Healthy *E. sinensis* specimens of a similar size and with complete appendages were collected from a Haorun Group breeding farm (Taizhou City, Jiangsu Province, China). The *E. sinensis* were fed with a feeding ratio of 5% at 9 am and 2 pm every day during the acclimation period. The daily culture management was as follows: the water was continuously aerated and renewed by half per day; residual feed and excrement were removed once a day; water quality was monitored each day (water temperature was maintained at (21 ± 1) °C; pH was at 7.8; the concentrations of NO_2_^−^ and NH_3_-N were 0.005 mg/L and 0.1 mg/L, respectively; the concentration of dissolved oxygen (DO) was 6 mg/L). Feeding was stopped the day before the experiment. In this study, five replicates of control and alkalinity groups were set up. Bicarbonate was added to achieve an alkalinity of 60 mmol/L. The total alkalinity was measured using the methyl orange hydrochloride calibration method. 

### 2.2. Proteomics Analysis of the Cerebral Ganglion of E. sinensis

#### 2.2.1. Sample Collection 

After 6 h of alkalinity treatment, six male and female *E. sinensis* were randomly collected from the control group, and six were randomly collected from the experimental group, and their cerebral ganglions were collected. The cerebral ganglions of a pair of male and female crabs from the same group were mixed as one sample. Finally, three cerebral ganglion samples were obtained from the control, and three were obtained from the experimental group. The samples were quickly frozen in liquid nitrogen and then stored at −80 °C for subsequent proteomics analysis.

#### 2.2.2. Proteomics Analysis

##### Protein Extraction, Separation, Enzymolysis, and Desalination

The tissues were homogenized in liquid nitrogen, and then a lysis solution and protease inhibitor were added to reach a final concentration of 1 mM; the mixture was then sonicated on ice. The solution was then centrifuged at room temperature for 10 min (12,000 rpm), and the supernatant was collected. It was then centrifuged again. The supernatant was the total protein solution of the sample. The protein concentration was determined using the bicinchoninic acid (BCA) method, and the samples were stored at −80 °C. Then, 10 mg of protein was taken from each sample and separated using 12% sodium dodecyl sulfate polyacrylamide gel electrophoresis (SDS-PAGE). The stained gel was scanned using an automatic digital gel imaging and analysis system. According to the protein concentration, an appropriate amount of protein was taken from each sample, and the samples in different groups were diluted and adjusted to the same concentration using lysis solution. The precipitate was collected, re-dissolved, and digested at 37 °C overnight. After digestion, the proteins were desalted in a SOLA™ SPE 96-well plate. After column activation, the samples were loaded and washed. Finally, 50% acetonitrile–water was used for elution.

##### Liquid Chromatography–Tandem Mass Spectrometry/Mass Spectrometry (LC-MS/MS) Analysis and Database Query

The samples were loaded onto a 25 cm C18 column (RP-C18, ionopticks) and eluted using a gradient with a flow rate of 300 nL/min. Mobile phase A: H_2_O-FA (99.9:0.1, *v*/*v*); mobile phase B: ACN-H_2_O-FA (80:19.9:0.1, *v*/*v*/*v*). The gradient elution procedures were as follows: 0–66 min, 3–27% B; 66–73 min, 27–46% B; 73–84 min, 46–100% B; 84–90 min, 100% B. The MS procedures were as follows: the capillary voltage was 1.4 KV; the drying air temperature was 180 °C; the drying gas flow rate was 3.0 L/min; the scanning range was 100–1700 *m*/*z*.

The LC-MS/MS original files were imported into MaxQuant (version 1.6.17.0) for database queries. The search engine Andromeda was used for the LFQ non-standard quantitative analysis. To prevent peak mismatch, the search standard was controlled by a false discovery rate (FDR) < 0.01. The specific database search parameters were set as follows: the missed cleavage was 2; the fixed modification was carbamidomethyl (C); the variable modification was oxidation (M); the decoy database pattern was reverse; the enzyme was trypsin; the first search peptide tolerance was 20 ppm; the main search peptide tolerance was 10 ppm.

##### Gene Ontology (GO) Annotation on Proteomics Analysis

Species proteins were used as the background list, and the total list of proteins obtained was used as the candidate list. A hypergeometric distribution test was used to calculate the *p*-value representing whether the functional set was significantly enriched in the differential protein list. Then, the *p*-value was corrected using multiple Benjamini–Hochberg tests to obtain the FDR. The proteins identified were annotated with Blast2GO [30].

##### Enrichment Analysis on Differentially Expressed Proteins (DEPs)

Log2(foldchange) (FC) was used to evaluate the expression change ratio of a protein between the experimental group and the control group. The *p*-value calculated by the t-test represented the significant difference. The screening conditions were FC > 1.5 or FC < 1/1.5 and *p*-value < 0.05 [31]. The top 20 Kyoto Encyclopedia of Genes and Genomes (KEGG) pathways were obtained according to the following conditions: the pathways in which the gene number in each pathway (listhit) was larger than one were selected, and then the top 20 KEGG pathways were obtained in descending order according to −log10*p-*value of the selected pathways.

### 2.3. Metabolomic Analysis of the Muscle of E. sinensis 

#### 2.3.1. Sample Collection 

After the 6 h alkalinity treatment, two *E. sinensis* (one female and one male) were randomly selected from each parallel group. The walking muscles were sampled. The muscle samples from one male and female *E. sinensis* from the same parallel group were mixed as one sample. Finally, five muscle samples were obtained and then quickly frozen in liquid nitrogen and stored at −80 °C for subsequent metabolomics analysis.

#### 2.3.2. Metabolomics Analysis 

##### Extraction of Metabolites from the Muscle of *E. sinensis* and Ultra Performance Liquid Chromatography/Mass Chromatography (UPLC-MS) Analysis 

For quality control (QC) samples, an equal volume of each sample was mixed and prepared for balancing using an LC/MS system; the status of the instruments and the stability of the system were monitored during the whole experiment process. Meanwhile, a blank sample was set to remove background interference. The extraction procedures were as follows: 100 mg samples were homogenized in liquid nitrogen and placed in Eppendorf (EP) tubes; then, 500 μL 80% methanol was added. The samples were placed in an ice bath for 5 min after vortexing and were then centrifuged at 4 °C. The supernatant was collected, diluted, and then centrifuged at 4 °C. Thereafter, the supernatant was collected and loaded for LC-MS analysis [32]. The samples were injected into Vanquish Ultra High-Performance Liquid Chromatography (UHPLC) (Thermo Fisher, Waltham, MA, USA). The column was Hypesil Goldcolumn (C18), with a column temperature of 40 °C and a flow rate of 0.2 mL/min. The electro-spray ionization (ESI) source settings were as follows: the spray voltage was 3.5 kV; the sheath gas flow rate was 35 psi; the aux gas flow rate was 10 L/min; the capillary temperature was 320 °C; the S-lens radio frequency (RF) level was 60; the aux gas heater temperature was 350 °C; the MS/MS second-level scanning was data-dependent scans.

##### Principal Component Analysis (PCA) and Partial Least Squares Discrimination Analysis (PLS-DA) on Differential Expressed Metabolites (DEMs)

PCA was used to analyze the overall distribution trends of samples between the control and experiment groups. PLS-DA was used to test whether the model was “over-fitting” [33]. The test procedures were as follows: group marks of the samples were randomly disrupted, and modeling and predictions were conducted. Each model corresponded to a set of R2 and Q2 values, and their regression lines could be obtained according to the Q2 and R2 values after 200 iterations of disruption and modeling. When the R2 was larger than the Q2, and the intercept between the Q2 regression line and the *Y*-axis was less than 0, it could be concluded that the model was not “over-fitting” [34].

##### KEGG Enrichment Analysis on DEMs 

The DEMs were screened according to the FC and *p*-value. Metabolites with FC > 1.5 or FC < 0.667 and *p* < 0.05 were selected as DEMs. The ratio condition was x/n > y/N, where N is the number of total metabolites participating in KEGG metabolic pathways; n is the number of DEMs in N; y is the number of metabolites annotated to a KEGG pathway; x is the number of DEMs enriched in a KEGG pathway. Pathways that met this ratio condition and had a *p*-value < 0.05 were set as significantly differentially expressed KEGG pathways and then the top 20 KEGG pathways were obtained.

### 2.4. Combined Proteomic and Metabolomic Analyses on Cerebral Ganglion and Muscle 

All of the DEPs and DEMs were mapped to the KEGG database, and then pathways in which DEPs and DEMs were both involved were determined as co-regulatory pathways in both omics.

## 3. Results

### 3.1. Proteomics Analysis 

#### 3.1.1. Basic Statistics and Annotation of Cerebral Ganglion Proteome 

##### Peptide Length and Molecular Weight Distribution 

The length statistics of the identified peptides are shown in Figure 1A. When the number of amino acids was 1–4, the number of peptides increased, and the number of peptides reached the maximum (635) when the number of amino acids reached four. The number of peptides then decreased as the number of amino acids increased when the number of amino acids was greater than four. The molecular weight distribution of the identified proteins is shown in Figure 1B. As shown in Figure 2, the molecular weights of the identified proteins were mainly distributed in the range of 1–120 kDa, with most being in the range of 10–60 kDa. The number of proteins with molecular weights greater than 200 kDa was 32.

##### GO Annotation of Proteome

As shown in Figure 2, the top 10 biological processes (BPs) mainly involved information processing and transport (No. 1, 2, 3, 5, and 6), environmental adaptation (No. 7), development (No. 8), and energy metabolism (No. 4, 9, and 10). The top 10 cellular components (CCs) mainly involved energy metabolism-relevant organelles (No. 2 and 4), the cytoskeleton (No. 3, 6, and 9), and the proteasome complex (No. 7 and 8). The top 10 molecular functions (MFs) were mainly related to energy metabolism (No. 1, 2, 4, 5, and 7), ion transportation (No. 3), and the cytoskeleton (No. 6, 9, and 10). 

#### 3.1.2. Enrichment Analysis of the Top 20 KEGG Pathways

As shown in Figure 3, the top 20 KEGG pathways could be mainly divided into four categories: signal transduction, amino acid metabolism, energy metabolism, and organismal system. Among them, six pathways were involved in signal transduction, which mainly involved circadian regulation, synaptic transmission, ion channels, and anti-stress response. Energy metabolism mostly involves carbohydrate metabolism. The organismal system mostly involves the regulation of the digestive, endocrine, and immune systems. 

### 3.2. Metabolomics Analysis 

#### 3.2.1. Multivariate Statistics of Differentially Expressed Metabolites 

##### PCA Analysis of Differential Metabolites

A three-dimensional principal component analysis was performed on the control group and the experimental group in order to verify the repeatability of the results. The results are shown in Figure 4. The first principal component (PC1) represented 21.65% of the characteristics of the raw data, the second principal component (PC2) accounted for 16.4% of the raw data characteristics, and the third principal component (PC3) accounted for 14.2% of the raw data characteristics. The aggregation of discrete points represented good similarity. In the present study, the PCA results showed significant separation between the experimental group and the control group, indicating significant changes in metabolites between these two groups.

##### PLS-DA Analysis of Differential Metabolites 

The results of the PLS-DA analysis are shown in Figure 5A. The principal component pattern separation between the experimental group and the control group was significant, as indicated by the fact that there was no overlapping between the distributions in the figure. The R2Y and Q2Y obtained by an analysis of differential metabolites were 0.99 and 0.85, respectively, indicating that the model could reflect 99% of the differences between the two groups. Moreover, the predictability of the differential metabolites of this model was 85%, and the R2Y and Q2Y were both higher than 0.5, indicating that the PLS-DA model was reliable. Permutation tests were carried out on the model parameters R2 and Q2, and the number of tests was set at 200. As shown in Figure 5B, the R2 values were all higher than the Q2 values, and Q2 < 0 indicated no over-fitting.

#### 3.2.2. Enrichment Analysis on Top 20 KEGG Pathways 

As shown in Figure 6, the top 20 KEGG pathways were mainly involved in four categories: energy, amino acid metabolism, signal transduction, and endocrine regulation. Among them, there were 11 terms relevant to energy metabolism (which was the most among the four categories), mainly involving arachidonic acid, TCA, nitrogen, carbon, and sulfur metabolism. Among them, there were nine upregulated pathways. Amino acid metabolism involved four pathways, including glutamate, taurine, histidine, and arginine, most of which were upregulated. Two pathways were related to signal transduction, involving synapse and phototransduction. The pathways that were associated with the endocrine system involved glucagon metabolism.

### 3.3. Combined Proteomics and Metabolomics Analyses of Cerebral Ganglion and Muscle of E. sinensis under Acute Alkalinity Stress 

#### 3.3.1. Co-Enrichment Pathways of Cerebral Ganglion Proteome and Muscle Metabolome

As shown in Figure 7, there were 11 pathways that were co-enriched in the cerebral ganglion proteome and muscle metabolome of *E. sinensis* under acute alkalinity stress, which were very similar to the enrichment results of the cerebral ganglion proteome and muscle metabolome when analyzed on their own. The co-enriched pathways mainly involved four categories: amino acid metabolism, energy metabolism, signal transduction, and organismal system. Among them, the four energy-metabolism-relevant pathways were the most enriched, which mainly included glycolipid metabolism. The signal transduction pathways mainly included synapse transduction and ion channel regulation. The organismal system mainly involves immunity regulation. The relevant DEPs and DEMs in the co-regulatory pathways are shown in Table 1.

#### 3.3.2. Synergistic Regulation Mechanisms of Cerebral Ganglion and Muscle of *E. sinensis* under Alkalinity Stress 

We carried out a comprehensive analysis of the synergistic regulation pathways and major DEPs and DEMs in both omics of *E. sinensis* under alkalinity stress. As shown in Figure 8, after NaHCO_3_ stress, the co-regulation pathways mainly involved four categories: amino acid metabolism, energy metabolism, signal transduction, and organismal system. They also included elements that played a regulatory role in five subcategories: the TCA cycle, nerve signal transduction, immune regulation, homeostasis maintenance, and ion transportation.

## 4. Discussion

We found that the top 20 KEGG pathways enriched from cerebral ganglion proteomics and muscle metabolomics included amino acid metabolism, energy metabolism, signal transduction, and organismal system and that a total of 11 pathways were co-enriched in both omics, which also involved the above-mentioned four categories, indicating that the cerebral ganglion and muscle of *E. sinensis* were closely related in the regulation of adaptive response to alkalinity stress.

### 4.1. Synergistic Regulation of Amino Acid Metabolism in Cerebral Ganglion and Muscle of E. sinensis under Alkalinity Stress

In this study, alanine, aspartate, and glutamate metabolism pathways exhibited an upregulated expression. Alanine is a product of aspartic acid metabolism. After a series of reactions, pantothenic acid and then pantothenate are produced, which can be used to synthesize coenzyme A [35]. This indicated that this pathway was also indirectly involved in energy metabolism. In this study, the expression of DEPs (aminobutyrate aminotransferase, ABAT) and DEMs, including alpha-ketoglutaric acid, citric acid, adenylosuccinic acid, and glutamic acid, in this pathway were all significantly upregulated.

As the most important inhibitory neurotransmitter in the central nervous system, the reductions or insufficiency in ABAT can lead to neuromotor delay, hyperreflexes, seizures, etc. [36]. A study on acclimation to low salinity in *Cynoglossus semilaevis* showed that ABAT also played an important regulatory role in maintaining osmotic pressure balance and salinity adaptation [37]. An analogous conclusion was also made in the present study, which demonstrated the anti-stress function of ABAT in *E. sinensis* under alkalinity treatment. Glutamic acid can deaminate and play an indispensable role in the disposal of excessive waste nitrogen. Furthermore, glutamate is a quick-responding excitatory neurotransmitter in the nervous systems of organisms, which plays a key regulatory role in synaptic plasticity [38]. Another study on the effects of acute salinity stress on *E. sinensis* showed that serum glutamate played an important regulatory role in maintaining osmotic balance [39]. Furthermore, glutamic acid is also an important amino acid for flavor. It has been reported that the muscle of *E. sinensis* fattened in saline–alkaline water exhibited higher glutamic acid content [13], which was consistent with the results of the present study. In this study, the upregulation of glutamate in the muscle of *E. sinensis* after alkalinity stress played an important regulatory role in cerebral ganglion signal transduction, osmotic regulation, and flavor improvement.

Alpha-ketoglutaric acid, an intermediate metabolite in the tricarboxylic acid cycle, is a key component in intracellular carbon and nitrogen metabolism, as well as a precursor of many important amino acids, such as glutamic acid. It has various biological functions, such as energy metabolism modulation and immuno-enhancement [40]. Research on the metabolomics of gills in rainbow trout under salinity stress has shown that alpha-ketoglutaric acid was closely related to salinity variation and that it played a key regulatory role in salinity adaptation [41]. In the present study, ketoglutaric acid in *E. sinensis* was also upregulated after alkalinity stress, which not only modulated amino acid metabolism but also played an important regulatory role in signal transduction and energy metabolism. Citric acid, another important intermediate in the tricarboxylic acid cycle, participates in energy metabolism. A study on the gill metabolomics of *Portunus trituberculatus* under low salinity stress indicated that the upregulation of citric acid could ensure adequate energy supply [42]. 

Adenylosuccinic acid, a metabolite in the purine nucleotide cycle, can induce the expression of the anti-stress factor Nrf2 and be beneficial for muscle damage mitigation. In addition, it is an intermediate in AMP synthesis, thereby participating in energy metabolism [43,44]. In the present study, after alkalinity stress, the upregulation of the above-mentioned DEPs and DEMs not only regulated amino acid metabolism but also performed functions in energy homeostasis maintenance, signal transduction regulation, and anti-stress response.

### 4.2. Synergistic Regulation of Energy Metabolism under Alkalinity Stress

After acute alkalinity stress, many pathways relevant to glucose and lipid energy metabolism were significantly enriched. Metabolites, including prostaglandin H2, prostaglandin B2, and leukotriene C4, in the arachidonic acid metabolism pathway, were significantly upregulated. Arachidonic acid, a widely distributed polyunsaturated fatty acid, is an important essential fatty acid that can generate prostaglandin, thromboxane, leukotriene, and other metabolites and plays an important role in organ function maintenance, inflammatory reactions, and other physiological processes. It has been reported that decreased prostaglandin caused by salinity resulted in skeletal abnormalities [45], which indicated the important regulatory role of prostaglandin in regulating lipid metabolism, osmotic response, and muscle tissue development. Research on the metabolism of the Pacific oyster (*Crassostrea gigas*) under salinity stress has suggested that C20 fatty acid, which is a precursor of prostaglandin, changed consistently with salinity. It mobilized prostaglandin and played a regulatory role in membrane lipid remodeling to allow for the organism to adapt to salinity, which reflected the important regulatory role of prostaglandin in adaptive responses under osmotic conditions [46]. 

Leukotrienes are a group of local hormones formed by arachidonic acid in the lipoxygenase metabolism pathway. They can cause the contraction of tracheal smooth muscle, stimulate vascular permeability, activate white blood cells, and mediate inflammatory response [47]. Lipoxin B4, unlike leukotrienes, is a potent anti-inflammatory arachidonic acid and counteracts the effects of pro-inflammatory arachidonic acid, which can inhibit the production of chemokines and inflammatory cytokines and promote non-inflammatory phagocytosis, thus promoting the rapid regression of inflammatory response [48]. Research on the gill metabolomics of *E. sinensis* under saline–alkaline stress has also indicated the upregulation of LXB4 after stress, which was consistent with the results of the present study [12]. Both of these studies revealed the extensive anti-inflammatory regulatory role of LXB4 in various tissues under different environmental conditions. In this study, leukotriene and LXB4 were both upregulated to function in immunity regulation and homeostasis maintenance.

The glyoxylate cycle is involved in the synthesis of two-carbon precursors, which enter the system as acetyl-coenzyme A. Research on the metabolome of crucian carp under alkalinity stress has revealed that the major differentially expressed pathways involved glyoxylate and dicarboxylate metabolism [49], which was consistent with the results of this study, indicating the important regulatory role of glyoxylate and dicarboxylate metabolism in energy metabolism under adverse environmental stress. Glycine, an important inhibitory neurotransmitter in the nervous system, is synthesized from serine by serine hydroxymethyltransferase (SHMT). It has been reported that it played an important regulatory role in the salinity adaptation of crops [50]. In this study, the upregulated expression of serine hydroxymethyltransferase demonstrated its critical regulatory role in promoting nerve signal transduction, energy supply, and adaptation response to alkalinity stress.

Extra cholesterol is converted to cholesterol esters via the catalyzing actions of acetyl-CoA acetyltransferase (ACAT1) and then integrated into lipid droplets [51]. In this study, the downregulated expression of acetyl-CoA acetyltransferase could have inhibited the formation of lipid droplets and promoted lipolysis for energy supply.

Butyric acid can be absorbed into mitochondrial cells and plays an important role in energy supply. Moreover, butyric acid is converted into ketones via incomplete oxidation, which can stimulate the brain, promote the growth of neurons, release brain-derived neurotrophic factors, and play a regulatory role in serotonin, the immune pathway, and neuroendocrine factors [52]. In this study, the upregulated expression of this pathway played a positive regulatory role in protecting the nervous system and energy supply.

In the present study, the pentose and glucuronate interconversion pathways were significantly differentially expressed after alkalinity stress. The conversion between pentose and glucuronate, an important metabolic process in organisms, plays an important role in maintaining energy balance and modulating blood glucose levels [53]. As an important component of this pathway and a key enzyme in the polyol pathway, aldose reductase can catalyze the conversion of glucose to sorbitol. Sorbitol cannot be passively diffused out of the lens, which generates an osmotic gradient and induces water diffusing into the lens. The resulting swelling and electrolyte imbalance lead to the formation of cataracts [54]. In the present study, its downregulated expression after alkalinity stress could have played a positive regulatory role in the visual system protection of *E. sinensis*.

### 4.3. Synergistic Regulation of Signal Transduction and Immunoregulation under Alkalinity Stress

In this study, some signaling pathways relevant to signal transduction represented significantly differential expression after alkalinity stress, such as the serotonergic synapse and calcium signaling pathway.

Serotonin, an important neurotransmitter, plays a key regulatory role in diverse brain activities, such as learning and memory, cognitive control, sensory processing, autonomic response, and movement [55]. It has been reported that environmental stressors, such as pH, could cause the significant expression of serotonergic pathways, thereby enhancing the regulation of signal transduction to promote adaptive responses [56]. In the present study, the synergistic upregulation of the serotonergic synapse pathway in the cerebral ganglion and muscle was beneficial for the regulation of signal transduction and homeostasis maintenance in *E. sinensis* after alkalinity stress.

Ca^2+^, a universal intracellular signal in all eukaryotes, plays a regulatory role in diverse physiological processes. Calcium-transporting ATPase, an important regulatory enzyme in the calcium signaling pathway, serves to couple the energy generated by the hydrolysis of ATP with the active transport of key ions across membranes. It has been reported that decreased intracellular calcium concentrations resulted in muscle relaxation [57]. In the present study, it was observed that *E. sinensis* exhibited restlessness, constant swimming, and climbing in the early stages of alkalinity stress, so the downregulation of the calcium pathway could relieve over-strained muscle contractions in the early stages of alkalinity stress.

The ryanodine receptor (RyR), a tetramer channel protein, plays an important regulatory role in the release of intracellular calcium ions and muscle contraction. In addition, it has been reported that the downregulation of the ryanodine receptor could induce Ca^2+^ flow through ion channels in plasma membranes, then inhibit Ca^2+^-ATPase in the endoplasmic reticulum and reduce the apoptosis of perineuronal glial cells [58]. In the present study, its downregulation not only played a regulatory role in muscle relaxation, together with calcium-transporting ATPase, but it also played a modulatory role in ion channel homeostasis and cell apoptosis.

In this study, some immune-relevant signaling pathways appeared to be significantly upregulated after stress, such as the Fc epsilon Receptor I signaling pathway. The FcεRI complex plays a regulatory role in the activation of mast cells and basophils and is involved in IgE-mediated antigen presentation. Activated mast cells release preformed granules containing histamine and heparin and then release lipid mediators, such as leukotrienes C4 (LTC4), prostaglandins, and cytokines, which then trigger an immune response [59]. It has been reported that 96 h acute saline–alkaline stress caused the upregulation of signaling pathways relevant to the immune response in *E. sinensis* [15], which was consistent with the results of the present study. These results revealed the agile and positive adaptive response of the immune system in the cerebral ganglion and muscle of *E. sinensis* in response to alkalinity stress.

## 5. Conclusions

In the present study, the cerebral ganglion and muscle of *E. sinensis* played a significant synergistic regulatory role in response to alkalinity stress. This study provides a theoretical reference for further investigation into the regulatory mechanisms of *Eriocheir sinensis* under alkalinity stress. Moreover, as demonstrated in this study, *E. sinensis* exhibited a remarkable response to alkalinity stress, thereby demonstrating its potential as a model organism for environmental assessment.

## Figures and Tables

**Figure 1 animals-14-02374-f001:**
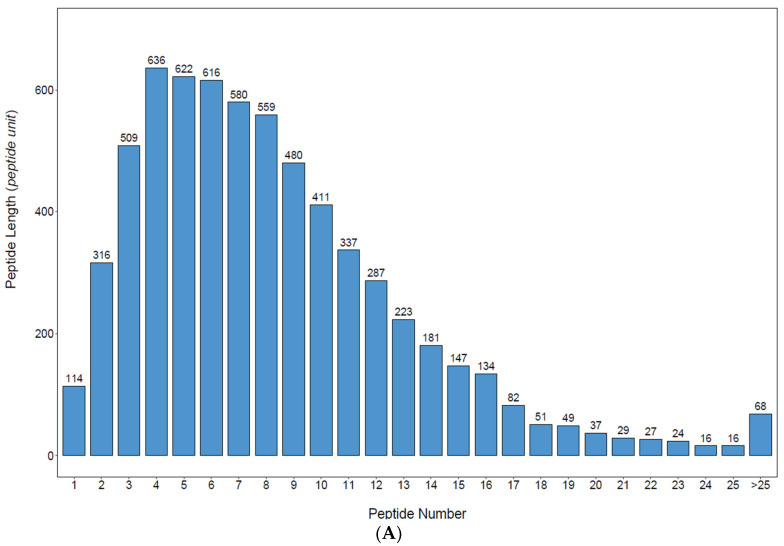
Basic statistics of the cerebral ganglion proteome of *Eriocheir sinensis*: (**A**) Peptide length distribution diagram (horizontal axis is the number of peptides of different amino acids and vertical axis is the length of the peptides); (**B**) Distribution of the molecular weight of proteins (horizontal axis is the molecular weights of the identified proteins and vertical axis is the number of proteins with different molecular weights).

**Figure 2 animals-14-02374-f002:**
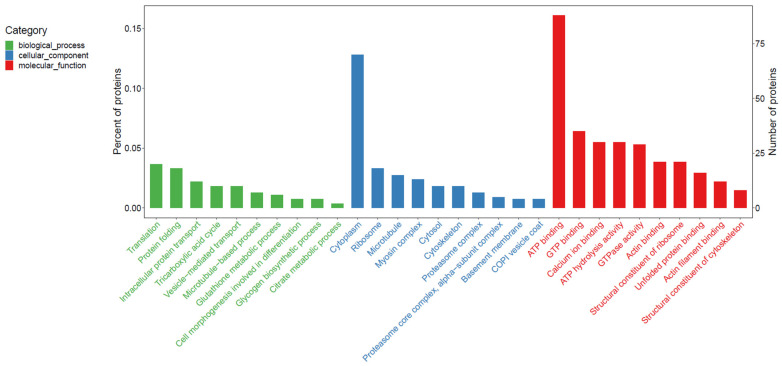
The GO entries in the figure are divided into three subcategories: biological processes (BPs), cellular components (CCs), and molecular functions (MFs). The horizontal coordinate indicates the GO item where the proteins were located, the left vertical coordinate indicates the proportion of protein in the different GO items, and the right vertical coordinate represents the number of proteins in each GO item.

**Figure 3 animals-14-02374-f003:**
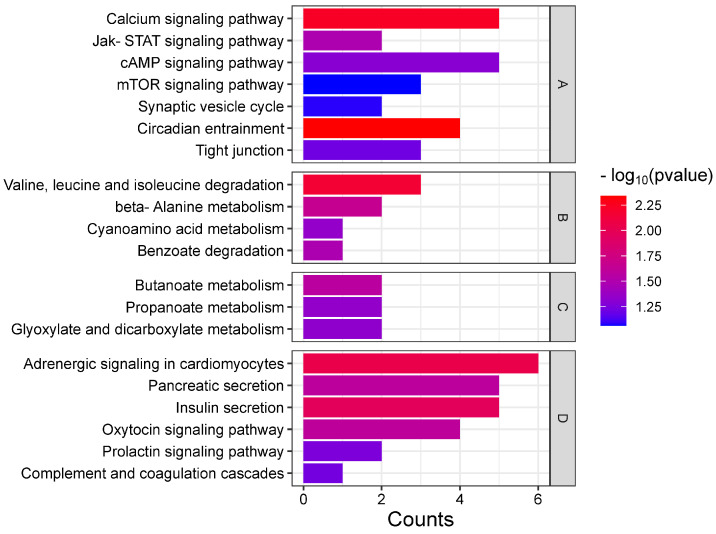
Top 20 KEGG pathways of cerebral ganglion proteomics under alkalinity stress. The abscissa indicates the number of proteins in each pathway, the vertical axis indicates the top 20 pathways, and the −log10*p-*value of each pathway is represented by different color. The four letters A–D represent the four functional categories that top 20 KEGG pathways can be classified into: A, signal transduction; B, amino acid metabolism; C, energy metabolism; D, organismal system.

**Figure 4 animals-14-02374-f004:**
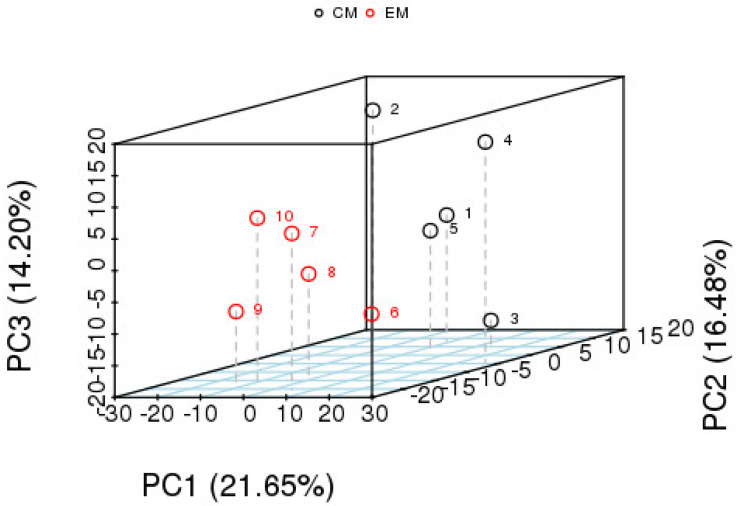
PCA analysis of differential metabolites of muscle metabolomics of *E. sinensis* under alkalinity stress.

**Figure 5 animals-14-02374-f005:**
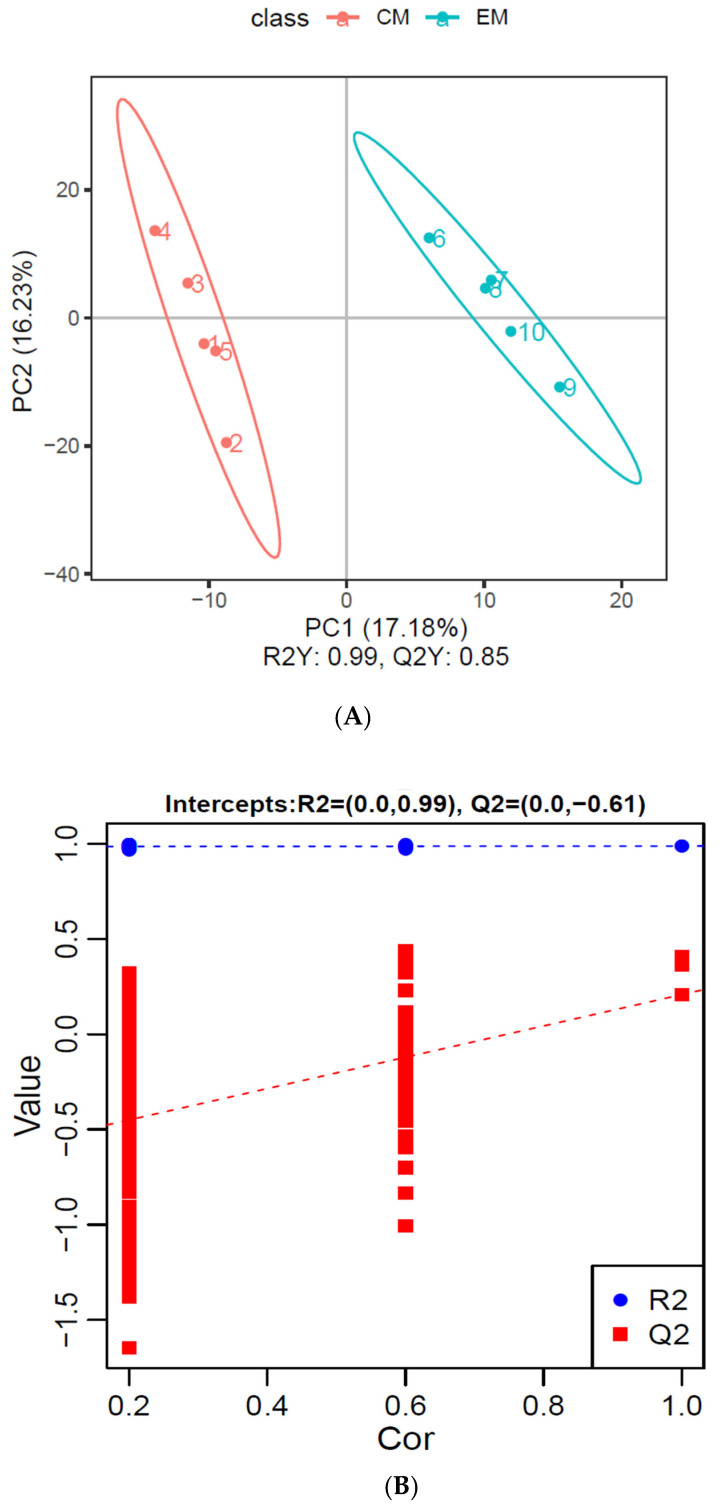
PLS−DA analysis of differential metabolites: (**A**) Analysis of PLS−DA; (**B**) Permutation test of PLS−DA.

**Figure 6 animals-14-02374-f006:**
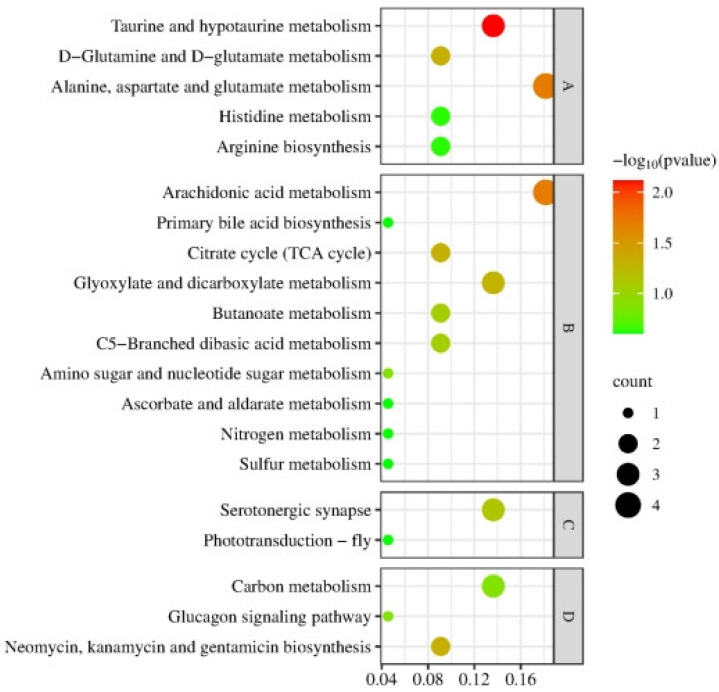
Top 20 KEGG pathways of differential metabolites in the muscle metabolome of *E. sinensis* under alkalinity stress. The horizontal axis represents the enrichment factor and the vertical axis indicates the pathway terms. The size of the points represents the number of metabolites in each pathway and the color of the points represents the −log10*p-*value of the pathways. A, amino acid metabolism; B, energy metabolism; C, signal transduction; D, endocrine regulation.

**Figure 7 animals-14-02374-f007:**
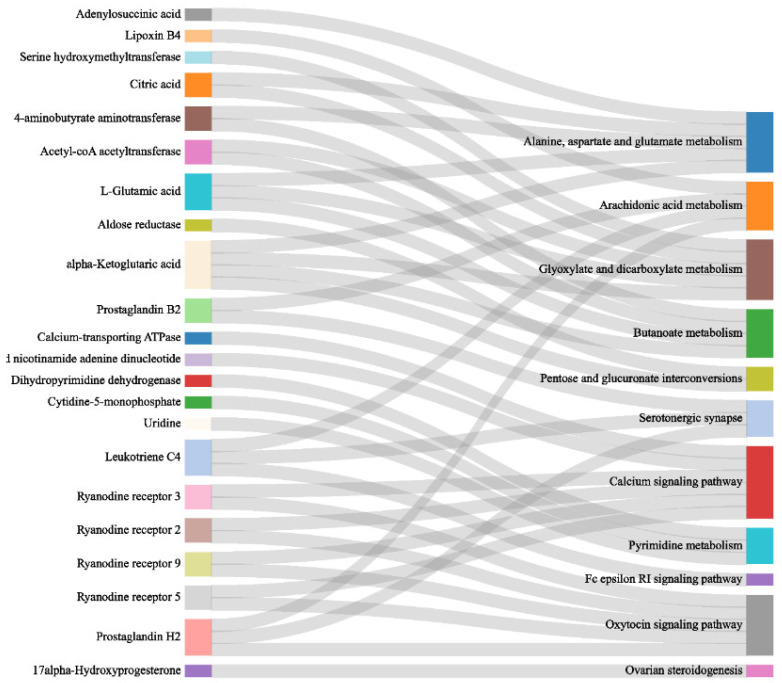
Co-regulatory pathways in both omics of *E. sinensis* under alkalinity stress. The rightmost axis represents the 11 co-enriched pathway terms and the leftmost axis represents the DEPs and DEMs corresponding to the top 11 pathways.

**Figure 8 animals-14-02374-f008:**
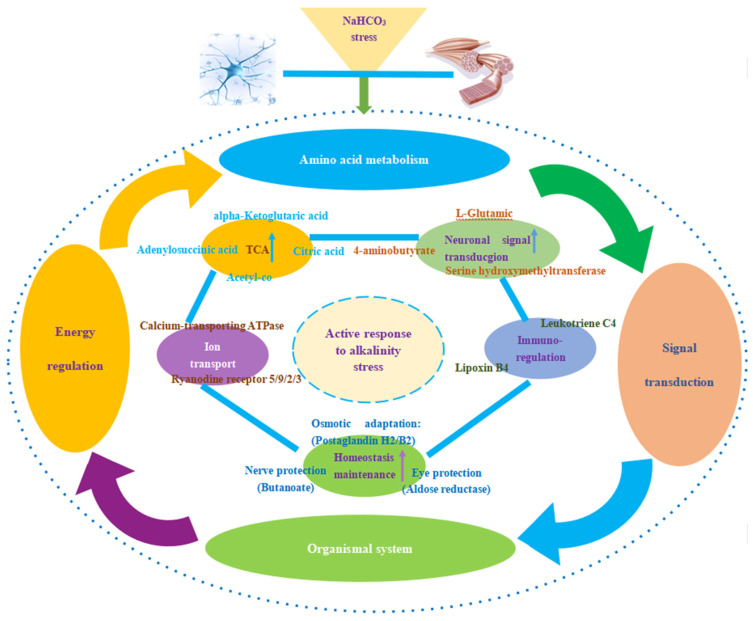
Schematic diagram of synergistic regulation mechanisms of cerebral ganglion and muscle of *E. sinensis* under alkalinity stress. The co-enrichment pathways of DEPs and DEMs involved four aspects: amino acid metabolism, energy metabolism, signal transduction, and organismal system. These were further divided into five functional groups: the TCA cycle, nerve signal transduction, immune regulation, homeostasis maintenance, and ion transport. The upper arrow represents the upregulated function group (all DEPs and DEMs were upregulated). The DEPs and DEMs in the five functional groups are marked beside them.

**Table 1 animals-14-02374-t001:** DEPs and DEMs in the co-regulatory pathways.

Name	Classification	Log2FC	*p*-Value	Regulation
4-aminobutyrate aminotransferase	DEP	inf	2.94 × 10−^6^	up
alpha-Ketoglutaric acid	DEM	1.81	4.90 × 10−^3^	up
Citric acid	DEM	0.73	4.28 × 10^−2^	up
Adenylosuccinic acid	DEM	0.99	2.95 × 10^−2^	up
L-Glutamic acid	DEM	0.71	3.63 × 10^−2^	up
Prostaglandin H2	DEM	1.51	3.79 × 10^−2^	up
Prostaglandin B2	DEM	1.67	1.29 × 10^−2^	up
Leukotriene C4	DEM	1.197	4.09 × 10^−2^	up
Lipoxin B4	DEM	1.52	1.50 × 10^−3^	up
ABAT	DEP	inf	2.94 × 10^−6^	up
Aldose reductase	DEP	0.2	4.00 × 10^−4^	up
Calcium-transporting ATPase	DEP	−0.39	2.87 × 10^−2^	down
Ryanodine receptor 5	DEP	−3.07	1.50 × 10^−2^	down
Ryanodine receptor 9	DEP	0.12	2.77 × 10^−3^	up
Ryanodine receptor 2	DEP	0.35	1.12 × 10^−3^	up
Ryanodine receptor 3	DEP	0.21	3.75 × 10^−4^	up
Reduced nicotinamide adenine dinucleotide	DEM	−5.25	5.60 × 10^−3^	down
Uridine	DEM	−0.54	1.04 × 10^−2^	down
Cytidine-5′-monophosphate		−4.81	8.90 × 10^−3^	down
17alpha-Hydroxyprogesterone	DEM	0.32	3.02 × 10^−3^	up
Serine hydroxymethyltransferase	DEP	0.63	5.72 × 10^−4^	up
Acetyl-coA acetyltransferase	DEP	0.98	1.07 × 10^−4^	up

## Data Availability

The raw data for proteomics and metabolomics in this study have been submitted to public database BIG Sub (PRJCA027286).

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
