# Peer review of "Exploration of Synergistic Regulation Mechanisms of Cerebral Ganglion and Muscle in Eriocheir sinensis Activated in Response to Alkalinity Stress"

_animals, 2024, doi:10.3390/ani14162374_

Round 1
Reviewer 1 Report
Comments and Suggestions for Authors
The present study provides a theoretical reference for further exploration on the growth and development mechanism of E. sinensis under alkaline environment, and also provided theoretical guidance for vigorous development of breeding industry of E. sinensis in saline-alkaline land and better development and utilization of global saline-alkaline land. The work is interesting for the scientific community and deals with a current problem, which is the acidification of waters caused by climate change. It serves both to alert the scientific community about climate change, and can also be used in strategies for aquaculture (this was not explored by the authors). The article needs to be substantially improved, there are formatting errors distributed throughout the work, including different font sizes throughout the template. English needs to be revised, as it is not fluent and makes reading difficult. The text is often tiring or does not contain citations in important sentences and paragraphs. The figures are of low quality and should be improved.
I also noticed that many paragraphs lack objectivity.
Title: It's very extensive. A compact title is required
Line 13-15: Why???
Abstract: Include the purpose of the article
Line 21: Delete “(1)”
Line 25: replace “;” with “.”
Line 25: Delete “(2) Methods:”
Line 26: replace “;” with “.”
Line 26: Delete “(3) Results:”
Line 30: replace “;” with “.”
Line 30: “(4) Conclusions:”
Line 35: Do not use words that are in the title of the work. Search for synonyms.
Line 38-40: Rewrite, it's confusing. You need to add a citation to this sentence.
Line 40-43: You need to add a citation to this sentence.
Line 43-45: What research? Cite articles that had these conclusions.
Line 46-47: You need to add a citation to this sentence.
Line 48: According to the incomplete statistics. Is this information reliable?
Line 48: UNESP and FAO are acronyms. Acronyms must be in parentheses when cited for the first time.
Line 48-51: In what year?
Line 56-57: It is highlighted by which authors?
Line 57-58: Cite articles that had these conclusions.
Line 60-62: Rewrite, it's confusing. There is a quote that is not numbered in the middle of the sentence and one also at the end.
Line 66: E, sinensis. Correct the scientific name.
Line 74-75: There are two scientific names that must be formatted in italics.
Line 79-81: Rewrite, it's confusing.
Line 87-88: You need to add a citation to this sentence.
Line 91-92: You need to add a citation to this sentence.
Line 92: Litopenaeus vannamei – in italic.
Line 117-128: Rewrite. You use words that are too short and they lose their meaning. Put the sentences together and give fluency to the topic
Line 140-158: There is a lot of detail about reagents. It doesn't make it attractive to the reader. I recommend cutting back and being more objective.
Line 197-217: There is a lot of detail about reagents. It doesn't make it attractive to the reader. I recommend cutting back and being more objective.
Figure 1: It is essential to include the letters A and B in the figures. Figure 1B is of poor quality. You have to standardize the formatting of the figures.
Line 350-357: I think part of the figure was over this paragraph.
Figure 8: There are some words outside the forms. It is necessary to organize.
Line 427: There's space left.
Line 470-494: Divide this paragraph into two paragraphs.
Line 563-574: It's still a result.
Conclusion: How will your work be important in assessing climate change and aquaculture? What are the future perspectives for this research? Answer this and you will have the conclusion.
Comments on the Quality of English LanguageEnglish needs to be revised, as it is not fluent and makes reading difficult.
Author Response
Dear reviewer,
Thank you very much for your comments concerning our manuscript entitled “ Exploration of synergistic regulation mechanism of cerebral ganglion and muscle in Eriocheir sinensis in response to alkalinity stress”. These comments are very helpful for revising and improving our manuscript. We have studied the comments carefully and have made major modification. This modified manuscript has been reviewed by an experienced English-speaking expert. We think this modified manuscript can reach the requirement of publication. The modification are marked in red. The response is as follows,
1.Title: It's very extensive. A compact title is required
Response: Thanks for your comment. We have modified the title to a compact one (Line 2-4).
- Line 13-15: Why???
Response: Thanks for your comment. We have modified the “Simple Summary” to make it more accurate and concise. (Line 12-18).
- Abstract: Include the purpose of the article
Response: Thanks for your comment. We have added the purpose of the article (Line 19-21).
- Line 21: Delete “(1)”
Line 25: replace “;” with “.”
Line 25: Delete “(2) Methods:”
Line 26: replace “;” with “.”
Line 26: Delete “(3) Results:”
Line 30: replace “;” with “.”
Line 30: “(4) Conclusions:”
Response: Thanks for the comment about format of Abstract. We have modified according to the above comments, we have deleted “(1)”, replaced “;” with “.”, deleted “(2) Methods:”, replaced “;” with “.”, deleted “(3) Results:”, replaced “;” with “.”. We have added conclusions (Line 19-29)
- Line 35: Do not use words that are in the title of the work. Search for synonyms.
Response: Thanks for your comment. We keep the necessary key words and replace the words not so important (Line 30).
- Line 38-40: Rewrite, it's confusing. You need to add a citation to this sentence.
Response: Thanks for your comment. We consider carefully and we think that this information is a common sense, it’s really not necessary to be described here. We have deleted it.
- Line 40-43: You need to add a citation to this sentence.
Response: Thanks for your comment. We have added citations here (Introduction Para.1 Line 36).
- Line 43-45: What research? Cite articles that had these conclusions.
Response: Thanks for your comment. We have modified the sentences and cited relevant references (Introduction Para 1 Line 36-38).
- Line 46-47: You need to add a citation to this sentence.
Response: Thanks for your comment. We think this sentence is not subjective, we have deleted it.
10.Line 48: According to the incomplete statistics. Is this information reliable?
Line 48: UNESP and FAO are acronyms. Acronyms must be in parentheses when cited for the first time.
Response: Thanks for your comment. I have checked this information and made relevant modification including the cited reference (Introduction Para.1 Line 38-40).
- Line 48-51: In what year?
Response: Thanks for your comment. The year has been added (Introduction Para.1 Line 39).
- Line 56-57: It is highlighted by which authors?
Response: Thanks for your comment. This is not subjective, we have deleted it.
- Line 57-58: Cite articles that had these conclusions.
Response: Thanks for your comment. The relevant references are indicated at the end of these conclusions (Introduction Para.1 Line 45).
- Line 60-62: Rewrite, it's confusing. There is a quote that is not numbered in the middle of the sentence and one also at the end.
Response: Thanks for your comment. We have rewritten this sentence (Introduction Para.1 Line 46-47).
- Line 66: E, sinensis. Correct the scientific name.
Response: Thanks for your comment. The sentence has been rewritten. (Introduction Para.1 Line 52).
- Line 74-75: There are two scientific names that must be formatted in italics.
Response: Thanks for your comment. These two scientific names have been modified (Introduction Para.2 Line 57-58).
- Line 79-81: Rewrite, it's confusing.
Response: Thanks for your comment. It has been rewritten (Introduction Para.2 Line 64-65)..
- Line 87-88: You need to add a citation to this sentence.
Response: Thanks for your comment. This sentence doesn’t have new information, we think carefully and deleted it.
- Line 91-92: You need to add a citation to this sentence.
Response: Thanks for your comment. This information is obvious, so we have deleted it.
- Line 92: Litopenaeus vannamei – in italic.
Response: Thanks for your comment. We have modified it ((Introduction Para.3 Line 75).
- Line 117-128: Rewrite. You use words that are too short and they lose their meaning. Put the sentences together and give fluency to the topic.
Response: Thanks for your comment. We have modified it (Materials and methods section 2.2 Line 97-107).
- Line 140-158: There is a lot of detail about reagents. It doesn't make it attractive to the reader. I recommend cutting back and being more objective.
Response: Thanks for your comment. We have deleted the detailed information (Materials and methods section 2.2 Line 97-107).
- Line 197-217: There is a lot of detail about reagents. It doesn't make it attractive to the reader. I recommend cutting back and being more objective.
Response: Thanks for your comment. We have deleted the unnecessary and detailed information (Materials and methods section 2.3.2.1 Line 118-131)..
- Figure 1: It is essential to include the letters A and B in the figures. Figure 1B is of poor quality. You have to standardize the formatting of the figures.
Response: Thanks for your comment. We have added the letters A and B in the figure 1 and replaced the figure 1B with a better one (Figure 1).
- Line 350-357: I think part of the figure was over this paragraph.
Response: Thanks for your comment. I have modified the figure.
- Figure 8: There are some words outside the forms. It is necessary to organize.
Response: Thanks for your comment. We want to explain about it . We especially put these words outside the forms to make full use of the space (Figure 8).
- Line 427: There's space left.
Response: Thanks for your comment. We have deleted the space (Discussion Para 1. Line 427).
- Line 470-494: Divide this paragraph into two paragraphs.
Response: Thanks for your comment. We have divided this paragraph into two (Discussion section 4.2 Line 474-502).
- Line 563-574: It's still a result.
Conclusion: How will your work be important in assessing climate change and aquaculture? What are the future perspectives for this research? Answer this and you will have the conclusion.
Response: Thanks for your comment. We have modified the conclusion and indicated the future perspectives for this research (Conclusion Line 576-581).
Reviewer 2 Report
Comments and Suggestions for Authors
The specific comments for this manuscript, are as follows,
1. Introduction
- In the first paragraph, author employed an entire paragraph to elucidate the peril of global warming, which could be condensed for conciseness.
- As far as I know, there’s one to two recent studies on alkalinity stress of Eriocheir sinensis which were not mentioned in this study, please incorporate these findings into the second paragraph.
2. Materials and methods:
- Overall, the description of “Materials and methods” was a little long. You may make some revision. For example, samples collected for proteome and metabolome analysis at the same time and the procedures can be described in the section 2.2. It’s unnecessary to describe the sampling for both omics separately for conciseness.
- Please provide statistical method for omics analysis.
3. Results
- Differential proteins and differential metabolites of the co-enriched pathways of both omics are important, please describe their detailed information in a separate table.
4. Discussion
- Information relevant to immune regulation mechanism is not much and not the main finding of this study, you may combine this paragraph (section 4.4) with section 4.3.
5. Conclusions
- The conclusions should be more concise and direct, the detailed description in brackets can be removed.
6. Others
- There are some formatting errors in the full text.
- For the subtitle of section 2.2(Line 116), “can” is redundant.
- On Line 81, the number “96” should be expressed in English words.
Author Response
Dear reviewer,
Thank you very much for your comments concerning our manuscript entitled “ Exploration of synergistic regulation mechanism of cerebral ganglion and muscle in Eriocheir sinensis in response to alkalinity stress”. These comments are very helpful for revising and improving our manuscript. We have studied the comments carefully and have made major modification. The modification are marked in red. The response is as follows,
- Introduction
In the first paragraph, author employed an entire paragraph to elucidate the peril of global warming, which could be condensed for conciseness.
Response: Thanks for your comment. We have compacted the first paragraph and deleted redundant information (Introduction Para.1 Line 33-53).
As far as I know, there’s one to two recent studies on alkalinity stress of Eriocheir sinensis which were not mentioned in this study, please incorporate these findings into the second paragraph.
Response: Thanks for your comment. We have modified it (Introduction Para.2 Line 50).
- Materials and methods:
- Overall, the description of “Materials and methods” was a little long. You may make some revision. For example, samples collected for proteome and metabolome analysis at the same time and the procedures can be described in the section 2.2. It’s unnecessary to describe the sampling for both omics separately for conciseness.
Response: Thanks for your comment. We have described the sampling for proteome and metabolome analysis together in the section 2.2 and deleted some detailed information (Materials and methods section 2.2 Line 97-107).
- Please provide statistical method for omics analysis.
Response: Thanks for your comment. According to your review, we checked and consulted the relevant technical personnel, statistical analysis of proteome and metabolome is not concentrated and it has been described dispersedly in the materials and methods. This is the reason that why it’s not so obvious (Materials and methods Line 117-210).
- Results
- Differential proteins and differential metabolites of the co-enriched pathways of both omics are important, please describe their detailed information in a separate table.
Response: Thanks for your comment. We have made a list for DEP and DEMs in Table 1 (Results section 3.3 Line 407).
- Discussion
- Information relevant to immune regulation mechanism is not much and not the main finding of this study, you may combine this paragraph (section 4.4) with section 4.3.
Response: Thanks for your comment. We have combined the section 4.4 with the section 4.3 and modified the subsection title of section 4.3 (Discussion section 4.3 Line 535-574).
- Conclusions
- The conclusions should be more concise and direct, the detailed description in brackets can be removed.
Response: Thanks for your comment. We have modified the conclusion and deleted the redundant detailed description to make it more concise and more meaningful (Conclusion Line 576-581).
- Others
- There are some formatting errors in the full text.
- For the subtitle of section 2.2(Line 116), “can” is redundant.
- On Line 81, the number “96” should be expressed in English words.
Response: Thanks for your comment. We have modified the formatting errors, we have deleted redundant word in the title of section 2.2 and modified the “96” into “Ninety-six” (Introduction Para 2 Line 66; Materials and methods section 2.2 Line 96).
Reviewer 3 Report
Comments and Suggestions for Authors
Please have an English teacher rewrite the paper and resubmit. Very difficult to read the English.

Please have an English teacher rewrite the paper and resubmit. Very difficult to read the English.
Author Response
Dear reviewer,
Thank you very much for your comments concerning our manuscript entitled “ Exploration of synergistic regulation mechanism of cerebral ganglion and muscle in Eriocheir sinensis in response to alkalinity stress”. These comments are very helpful for revising and improving our manuscript. We have studied the comments carefully and have made major modification. This modified manuscript has been reviewed by an relevant experienced English-speaking expert. We think this modified manuscript can reach the requirement of publication. The modification are marked in red. The response is as follows,
- Response: Thanks for your comment. According to the comments you marked on the manuscript, we have added the full names for the abbreviated words, and some improper words and sentences have been modified (The whole manuscript, Line 122, 124, 132, 147, 155, 158, 172, 174, 178, 182, 1874, 189 and 565).
- Explain potential differences between male and female crab ganglion responses. Mixing them together may produce inaccuracy in the results.
Response: Thanks very much for your comment. The specific response mechanism of female and male crab can be indicated using female or male crab separately. Actually, mixing samples and make relevant analysis are a common method to make some general research. In the future, we can make relevant research with female or male crabs, respectively.
- You marked some words with the first letter capitalized in the middle of the sentence means that the first letter of them shouldn’t be capitalized.
Response: Thanks for your comment. These are the names of pathways, so we think the first letter of them should be capitalized (Discussion Line 431, 506, 507 and 525). We have confirmed the commonly used abbreviation of Ryanodine receptor, that is RyR (Discussion Line 556).
- For conclusions, how can this information help to improve crab farming and crab health in a saline-alkaline system? Have you found the alkalinity level where the crabs fail to osmoregulate and then die ?
Response: Thanks for your comment. At present, there’s very few information about mechanism of alkalinity stress in Eriocheir sinensis, this study provides more theoretical information on response mechanism of E. sinensis under alkalinity stress. Furthermore, this manuscript indicated that there’s synergistic regulation between cerebral ganglion and muscle, so we need to consider neural regulation when you want to improve the muscle quality. We have modified the whole conclusion (Line 576-581). It has been reported the acute toxic stress of E. sinensis under alkalinity stress, please refer to this reference (Yang Yuhong et al. Study on toxicity of salinity and alkalinity on Eriocheir sinensis. Journal of Northeast Agricultural University, 2021, 53(2): 36-41. DOI:10.19720/j.cnki.issn.1005-9369.2022.02.005.)
- Response: Thanks for your comment. This modified manuscript has been reviewed by an experienced English-speaking expert, it can reach the requirement of publication.
Round 2
Reviewer 1 Report
Comments and Suggestions for Authors
When evaluating the manuscript, I was able to observe a significant revision.
I still suggest some adjustments:
1) Include the measurement unit in Figure 1A, y axis - peptide length. What is the unit of measurement? put in parentheses. "Peptide length (???)
2) Figure 1B. Standardize figure 1a equal to 1b. Note that figure 1a has a y axis and 1b does not. Standardize the figures.
Line 279. It has overlapping words.
Figure 6: There is a part below that needs to be formatted.
Author Response
1) Include the measurement unit in Figure 1A, y axis - peptide length. What is the unit of measurement? put in parentheses. "Peptide length (???)
Reponse: Peptide length unit has been added. (figure1A)
2) Figure 1B. Standardize figure 1a equal to 1b. Note that figure 1a has a y axis and 1b does not. Standardize the figures.
Line 279. It has overlapping words.
Figure 6: There is a part below that needs to be formatted.
Response: Figure 1 A and 1B has been modified , the overlapping words has been shown completely, part below figure 6 has been modified (figure 1A and 1B).
Reviewer 3 Report
Comments and Suggestions for Authors
Authors completed major English revisions and responded satisfactorily to issues raised.
Author Response
Thanks very much for your review and valuable comments.